# Brain and behavioural anomalies caused by *Tbx1* haploinsufficiency are corrected by vitamin B12

Marianna Caterino[3,4,*], Debora Paris[2,*], Giulia Torromino[5,9,*], Michele Costanzo[3], Gemma Flore[1], Annabella Tramice[2], Elisabetta Golini[5], Silvia Mandillo[5], Diletta Cavezza[5,6,7], Claudia Angelini[8], Margherita Ruoppolo[3,4], Andrea Motta[2], Elvira De Leonibus[5,6], Antonio Baldini[3], Elizabeth Illingworth[10], Gabriella Lania[1]

The brain-related phenotypes observed in 22q11.2 deletion syndrome (DS) patients are highly variable, and their origin is poorly understood. Changes in brain metabolism might contribute to these phenotypes, as many of the deleted genes are involved in metabolic processes, but this is unknown. This study shows for the first time that *Tbx1* haploinsufficiency causes brain metabolic imbalance. We studied two mouse models of 22q11.2DS using mass spectrometry, nuclear magnetic resonance spectroscopy, and transcriptomics. We found that *Tbx1*[+/−] mice and *Df1*/[+] mice, with a multigenic deletion that includes *Tbx1,* have elevated brain methylmalonic acid, which is highly brain-toxic. Focusing on *Tbx1* mutants, we found that they also have a more general brain metabolomic imbalance that affects key metabolic pathways, such as glutamine–glutamate and fatty acid metabolism. We provide transcriptomic evidence of a genotype–vitamin B12 treatment interaction. In addition, vitamin B12 treatment rescued a behavioural anomaly in *Tbx1*[+/−] mice. Further studies will be required to establish whether the specific metabolites affected by *Tbx1* haploinsufficiency are potential biomarkers of brain disease status in 22q11.2DS patients.

## Introduction

22q11.2 deletion syndrome is associated with a broad range of brain-related clinical signs and symptoms that is highly variable between patients, even within the same family. This includes structural brain anomalies, motor and cognitive deficits, learning difficulties, and a greatly increased risk for psychiatric disorders, especially anxiety and schizophrenia (1). The contribution of individual deleted genes to the brain phenotypes is largely unknown, or unproven, because of the rarity of point mutations. However, rare point mutations in the *TBX1* gene indicate that *TBX1* haploinsufficiency is responsible for most of the physical abnormalities associated with 22q11.2DS, and they have also been associated with attention deficits, mild mental retardation, learning difficulties, developmental delay, Asperger's syndrome, and depression (2, 3, 4, 5). However, the different nature of the brain phenotypes observed in 22q11.2DS patients suggests that additional genes from the deleted region likely play a role.

Intriguingly, clinical studies suggest that 22q11.2DS patients might have altered brain metabolism (6, 7, 8). In particular, a study of the metabolome and of mitochondrial function in a small group of children with 22q11.2DS revealed a distinct biochemical profile, which was consistent with increased oxidative stress, and shared features with congenital propionic and/or methylmalonic acidaemia (6).

Given the high genetic and phenotypic relevance of the available mouse models of 22q11.2DS, they provide, potentially, a unique opportunity to corroborate newly identified phenotypes in patients and to study the pathogenetic mechanisms underlying the brain-related deficits. Many orthologs of the del22q11.2 genes are expressed in the mouse brain (9), and at least nine orthologs are involved in mitochondrial metabolism, namely, *SLC25A1*, *TXNRD2*, *MRPL40*, *PRODH*, and *COMT*, all of which are expressed in mitochondria, and *TANGO2*, *ZDHHC8*, *UFD1L*, and *DGCR8*, all of which impact mitochondrial function (10, 11, 12). Thus, combined heterozygosity of multiple mitochondrial genes might negatively impact brain development and/or brain function, especially given the high energy demands of the brain.

We reasoned that mouse models of 22q11.2DS might have metabolic disturbances similar to those reported in patients, in which case the in vivo genetic and pharmacological manipulation

[1]Institute of Genetics and Biophysics of the National Research Council, Naples, Italy  [2]Institute of Biomolecular Chemistry, National Research Council, Pozzuoli (Naples), Italy  [3]Department of Molecular Medicine and Medical Biotechnology, University of Naples "Federico II", Naples, Italy  [4]CEINGE-Biotecnologie Avanzate Franco Salvatore, Naples, Italy  [5]Institute of Biochemistry and Cell Biology (IBBC), National Research Council (CNR), Monterotondo (Rome), Italy  [6]Telethon Institute of Genetics and Medicine, Pozzuoli (Naples), Italy  [7]PhD Program in Behavioural Neuroscience, Sapienza University of Rome, Rome, Italy  [8]Institute for Applied Mathematics "Mauro Picone," National Research Council, Naples, Italy  [9]Department of Humanistic Studies, University of Naples Federico II, Naples, Italy  [10]Department of Chemistry and Biology University of Salerno, Fisciano, Italy

Correspondence: eillingworth@unisa.it; gabriella.lania@igb.cnr.it
*Marianna Caterino, Debora Paris, and Giulia Torromino contributed equally to this work

of the model should provide insights into the contribution of the metabolic phenotypes to the disease pathogenesis and/or disease mechanisms. Indeed, we have previously shown that vitamin B12 (vB12, cyanocobalamin) treatment ameliorates some of the heart and brain abnormalities observed in *Tbx1* mutant embryos and adult mice ([13], [14], [15]), suggesting that it acts downstream of Tbx1.

vB12 acts as a cofactor in the methionine cycle for the production of reaction intermediates such as SAM (S-adenosyl methionine), which in turn provides methyl groups for the methylation of proteins and nucleic acids. In addition, in the mitochondria, it is a cofactor of methylmalonyl-CoA mutase, the enzyme that catalyses the reversible isomerization of methylmalonyl-CoA to succinyl-CoA, an essential intermediate in the citric acid cycle (TCA, Krebs cycle). Alterations of this pathway in humans and mice can lead to an accumulation of methylmalonic acid (MMA), which is highly brain-toxic ([16], [17], [18]).

In this study, we applied a multidisciplinary approach involving metabolomic, transcriptional, and behavioural studies to search for metabolic alterations in the brain of adult *Tbx1* heterozygous mice. Through this approach, we identified a new metabolic phenotype that was associated with reduced prepulse inhibition (PPI) of the acoustic startle response. We also discovered that the metabolic and behavioural phenotypes responded to vB12 treatment. In particular, the biochemical analysis revealed the accumulation of MMA and alteration of metabolites in vB12-related pathways in the brain of adult $Tbx1^{+/-}$ mice, and in mice carrying a multigenic deletion ($Df1/^+$) of 22q11.2DS orthologs that includes *Tbx1* ([19]).

Overall, our results show that *Tbx1* haploinsufficiency is associated with significant metabolic abnormalities in the young adult brain. We found that postnatal vB12 treatment corrected some of the metabolic alterations and it eradicated the PPI deficits.

# Results

## MMA accumulates in the brain of *Tbx1* and *Df1* heterozygous mutant mice

We were intrigued by the possibility that patients with 22q11.2DS have altered levels of brain-toxic metabolites related to the vB12 pathway. Human studies of brain metabolism have mainly been conducted on children and adolescent patients. Therefore, to determine whether similar anomalies were present in the mouse models, we performed our studies on young mice between 1 and 2 mo of age ([20]). A scheme of the experimental groups used in this study is shown in Fig S1C.

We first conducted a targeted metabolite assay using isolated whole brains of male and female $Tbx1^{+/-}$ and WT mice (Table S1, Group 1 in Fig S1C). A set of metabolites was quantified in brain extracts by liquid chromatography–tandem mass spectrometry (LC-MS/MS) (Table S1). The significant quantitative metabolic alterations in $Tbx1^{+/-}$ mice were provided by univariate analysis, as shown in the volcano plot (Fig 1A). The metabolites showing significant differences in $Tbx1^{+/-}$ versus WT were MMA, which was twofold higher in $Tbx1^{+/-}$ brains compared with WT brains (Fig 1A and B), and lactic acid (LA) (Fig 1A).

The human deletion is better modelled by $Df1/^+$ mice, which carry a heterozygous deletion of 27 orthologs of del22q11.2 genes, 9 of which are involved in metabolic processes, including 5 that are involved in mitochondrial activity. We analysed the whole brain tissue isolated from $Df1/^+$ and WT (control) male mice (n = 6 per genotype, Table S1, Group 2 in Fig S1C). Results showed that the mean concentration of MMA was threefold higher in $Df1/^+$ brains compared with WT brains (Figs 3 and S1A), $Df1/^+$ (PBS). In addition, C16 (hexadecanoylcarnitine), C16:1OH (3-hydroxyhexadecenoylcarnitine), C12:1 (dodecenoylcarnitine), C5 (valerylcarnitine), C5OH (3-hydroxyisovalerylcarnitine/3-hydroxy-2-methylbutyrylcarnitine), C18 (octadecanoylcarnitine), and C10:1 (decenoylcarnitine) were increased in the $Df1/^+$ brain; conversely, C6DC (methylglutarylcarnitine) was reduced (Fig S1B and Table S1).

Interestingly, we did not observe any significant variations in MMA concentration in non-CNS tissues of adult male and female $Tbx1^{+/-}$ mice (Fig S2), indicating that this phenotype is brain-specific. Moreover, MMA was undetectable in the brain of preterm WT and $Tbx1^{+/-}$ embryos (n = 5 for genotype). Together, these findings suggest that *Tbx1* haploinsufficiency affects mitochondrial metabolic processes related to vB12 specifically during postnatal and adult life.

## *Tbx1* haploinsufficiency alters the brain metabolomic profile of young adult mice

To visualize the effects of *Tbx1* haploinsufficiency on cell metabolism across the genome, we evaluated the metabolomic and transcriptional profiles of the brains of adult $Tbx1^{+/-}$ and WT mice (Group 3 in Fig S1C, Table S1) that had been treated with vB12 or with PBS (vehicle) twice a week for 4 wk starting at 4 wk of age.

At the end of treatment, the animals were euthanized, the brains surgically removed, and the two hemispheres separated. One hemisphere was used for NMR analysis, whereas the other was used for whole-genome transcriptomic analysis by RNA QuanTiseq. Each brain hemisphere was treated separately for the NMR analysis and for RNA-seq analysis, without pooling samples, and each dataset was treated as a biological replicate.

The effects of the *Tbx1* mutation on brain metabolism were analysed first by reporting the results obtained in PBS-treated $Tbx1^{+/-}$ and WT brains, followed by those obtained in vB12-treated animals with the same genotypes. To avoid potential variations because of sex differences, we used only male mice. For the NMR study, data from each brain hemisphere were obtained from 1D and 2D NMR experiments. Hydrophilic and lipophilic metabolites were assigned to spectral signals by comparing the observed chemical shift with published data ([21]) and/or online databases ([22]). To obtain the biochemical information from the NMR data, the 1D metabolic profiles underwent multivariate data analysis and projection methods. We applied an unsupervised principal component analysis (PCA) to assess class (genotype and/or treatment) homogeneity and to identify outliers, and then, we used a supervised OPLS-DA (orthogonal partial least squares discriminant analysis) to evaluate class discrimination and highlight metabolic differences associated with *Tbx1* mutation. The statistical analysis of the hydrophilic fraction (Fig 2A and A') yielded a regression model with high-quality parameters ($R^2$ = 0.99 [goodness of fit] and $Q^2$ = 0.94 [power in goodness of prediction] ([23]), CV-ANOVA test, $P$ = 0.0006), and a clear class

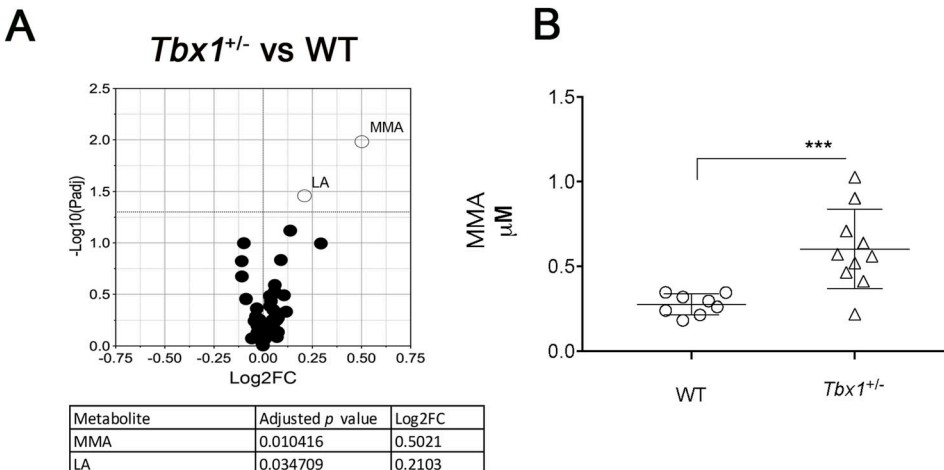

**Figure 1.  Targeted metabolome analysis by liquid chromatography–tandem mass spectrometry (LC-MS/MS).**
**(A)** Volcano plots showing differential concentrations of selected metabolites in (A) $Tbx1^{+/-}$ versus WT samples. The white dots represent the significant differential metabolites, and the black dots, all the metabolites identified in the dataset, the relative abundance of which was not significantly different between the groups. The differential metabolites are listed in the table accompanied by their corresponding values of difference (FC, fold change) and adjusted $P$-values. **(B)** Abundance of MMA ($\mu$M, mean ± SEM) in WT and $Tbx1^{+/-}$ brains. Differences between groups were evaluated by performing the Mann–Whitney test (*$P$ < 0.01, **$P$ < 0.005, ***$P$ < 0.0005, ****$P$ < 0.0001). Each symbol in the plot represents an individual animal.

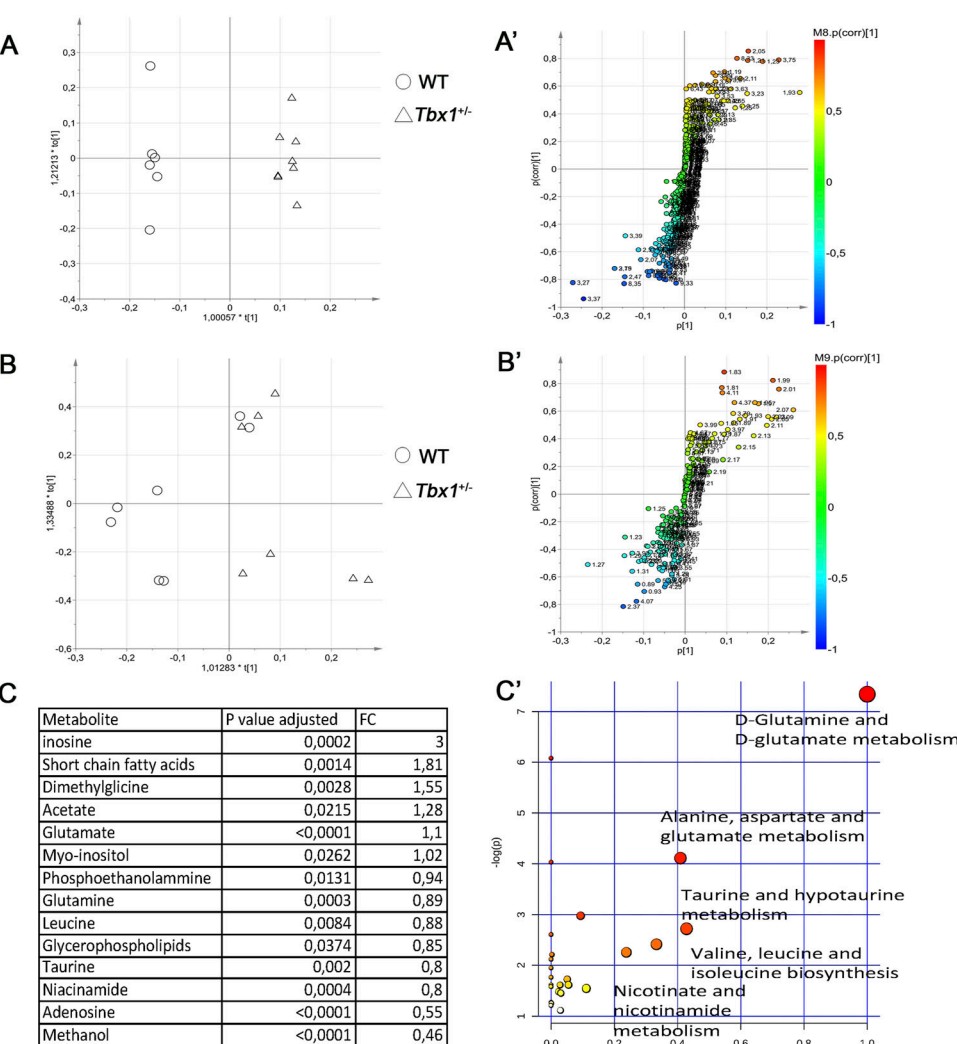

**Figure 2.  Brain metabolic profiles of $Tbx1^{+/-}$ and WT brains.**
**(A, B)** Score plot obtained from the orthogonal partial least squares discriminant analysis of brain extracts. **(A, B)** NMR data from hydrophilic (A) and hydrophobic (B) phases. The principal components t (1) and to (1) describe the space in which the NMR metabolic profiles of each brain extract are projected. Spectra of WT and $Tbx1^{+/-}$ samples are separated along the predictive t (1) axis (x-axis). **(A', B')** S plots associated with the multivariate model providing principal component visualization to facilitate model interpretation. The NMR variables situated far out on the wings of the S plot, in the lower left corner, indicate metabolites with an increased concentration in the WT class, whereas variables in the upper right side of the plot indicate metabolites with an increased concentration in $Tbx1^{+/-}$ mutant brain. The colour code refers to the correlation values. **(C)** Differential metabolites are listed in the table accompanied by their corresponding values of difference (FC, fold changes) and adjusted $P$-values. **(C')** Discriminating metabolites that were used as input for the pathway analysis in *Mus musculus* libraries to identify the most relevant pathways affected by the $Tbx1^{+/-}$ mutation. Pathway impact on the axis represents a combination of the centrality and pathway enrichment results; higher impact values on the x- and y-axes indicate the relative importance of the pathway; the size of the dots indicates how many metabolites within the pathway are altered, whereas the colour represents the significance (the more intense the red colour, the lower the $P$-value).

discrimination (Fig 2A). The WT class (empty dots) was located at negative t (1) values, whereas the $Tbx1^{+/-}$ class (empty triangles) was located at positive values. The discrimination along the t (1) parallel component (y-axis) accounted for the genotype separation, whereas the orthogonal component t (1) (x-axis) accounted for the intraclass homogeneity (Fig 2A). The associated S plot (Fig 2A') indicated the NMR chemical shifts, then assigned to metabolites, that discriminated between the two classes (Figs 2C and S3).

The statistical analysis of the lipophilic fraction (Fig 2B and B') generated a weak OPLS-DA model with only limited class separation (with parameters $R^2$ = 0.56 and $Q^2$ = 0.28, CV-ANOVA test, $P$ = 0.6), with a single metabolite (glycerophospholipids–sphingomyelin) responsible for the class distinction (Figs 2B, B', and C and S3). Together, the hydrophilic and lipophilic results revealed a group of compounds that characterized the brain metabolic differences between $Tbx1^{+/-}$ and WT mice (Fig 2A–C). In particular, for the hydrophilic fraction, we found several metabolites to be significantly different between $Tbx1^{+/-}$ and WT mice. Specifically, in $Tbx1^{+/-}$ mice, we observed downregulation of adenosine, niacinamide (vitamin B3), methanol, phosphoethanolamine, glutamine, leucine, glycerophospholipids–sphingomyelin (from the lipophilic fraction), and taurine, and upregulation of inosine, short-chain fatty acids, acetate, glutamate, myoinositol, and dimethylglycine (Figs 2C and S3, Table S1). To understand better the biological relevance of the data, we investigated the metabolic pathways in which the differently regulated molecular species are involved using MetaboAnalyst 4.0 software (https://www.metaboanalyst.ca/). The pathways found are depicted in Fig 2C'.

Using the discriminating metabolites and considering only the pathways with an impact of >40%, we found glutamine and glutamate metabolism ($P$ = 7.65 × 10$^{-4}$, impact 100%); alanine, aspartate, and glutamate metabolism ($P$ = 1.91 × 10$^{-2}$, impact 41%); and taurine and hypotaurine metabolism ($P$ = 7.13 × 10$^{-2}$, impact 42%) to be potentially altered in $Tbx1^{+/-}$ brains.

### $Tbx1$ haploinsufficiency has no significant effect on the brain transcriptome

We next sought to identify the transcriptional changes associated with the metabolic alterations identified in $Tbx1^{+/-}$ mice. We performed whole brain tissue RNA QuanTiseq analysis using the contralateral brain hemispheres to those used for NMR, and as for NMR, we performed the sequencing on individual hemispheres without pooling samples. The results showed that there were very few differentially expressed genes (DEGs) in $Tbx1^{+/-}$ versus WT brains (n = 3 out of 15,392 expressed genes, Mylk3, 4933413G19Rik, and Mfap5) (Fig S4D and Table S2). This surprising result might be due to a dilution effect on the target tissue (brain endothelial cells); $Tbx1$ is only expressed in a subset of brain vessels (~1.5% of brain cells) in the adult mouse (24), but it excludes significant non–cell-autonomous transcriptional effects on the brain parenchyma.

### Vitamin B12 treatment strongly affects the brain metabolome and transcriptome, and partially compensates for the effects of $Tbx1$ haploinsufficiency

Given the ability of vB12 prenatal treatment to rescue diverse phenotypes caused by $Tbx1$ haploinsufficiency (13, 14, 15), we asked

whether this occurs through metabolic mechanisms that are transcriptionally regulated by TBX1 or by rebalancing the brain metabolic profile of $Tbx1^{+/-}$ mice. To investigate the involvement of the vB12 pathway in the metabolic phenotype, we treated $Tbx1^{+/-}$ and $Df1/^+$ mice postnatally with vB12 or PBS (vehicle) twice a week for 4 wk starting at 4 wk of age, after which the animals were euthanized and the brains collected. We measured MMA concentration by LC-MS/MS in whole brain extracts (Group 2 in Fig S1C, Table S1). The results confirmed that $Tbx1^{+/-}$ and $Df1/^+$ brains have increased concentrations of MMA compared with WT controls (all animals were PBS-treated), and they showed that although vB12 treatment had no impact on MMA in WT animals, it was restored to WT levels in both of these mutants (Fig 3).

Next, we evaluated the brain metabolomic and transcriptional profiles of $Tbx1^{+/-}$ and WT mice treated postnatally with vB12 or PBS as described above (Group 3 in Fig S1C, Tables S1 and S2). NMR analysis and whole-genome transcriptomic analysis were performed on single brain hemispheres. In order to determine whether the metabolic changes observed in $Tbx1^{+/-}$ mice were related to the vB12 pathway, we applied NMR-based metabolomic analysis to the four classes (treatments and genotypes), that is, PBS-treated WT and $Tbx1^{+/-}$ mice and vB12-treated WT and $Tbx1^{+/-}$ mice. The OPLS-DA yielded a regression model with high-quality parameters (R2 = 0.93 and Q2 = 0.71, CV-ANOVA test, $P$ = 0.00055). The four classes were discriminated along three axes (Figs 4A and S5). As shown in the score plots in Fig 4A and in Fig S5A, the greatest impact on the metabolic profile was found in $Tbx1^{+/-}$ mice treated with vB12, revealed by the distance between this class (filled triangles) compared with the PBS-treated classes: WT (empty dots) and $Tbx1^{+/-}$ (empty triangles) along the first component t (1) axis (x-axis). The second component t (2) clearly showed the minor impact of vB12 on WT compared with the $Tbx1^{+/-}$ (compare filled dots with filled triangles). Instead, the third component t (3) axis (y-axis) revealed the difference between the PBS-treated $Tbx1^{+/-}$ and WT classes (compare empty dots with empty triangles). By assigning metabolites to the signals shown in the loading plot (Figs 4B and S5B) and quantifying the variation observed for each metabolite among the different classes, we found that overall, vB12 treatment induced significant metabolic changes in $Tbx1^{+/-}$ mice only (Figs 4A, S5A, and S6, Table S1, Group 3). In contrast, its impact on WT mice was very mild. In fact, the treatment induced changes in only six metabolites that were independent of the genotype, namely, citrate, acetate, aspartate, ethanolamine, formate, and fumarate (Fig S6). In only three cases did vB12 treatment rebalance a metabolic alteration observed in untreated $Tbx1^{+/-}$ mice, namely, inosine, glutamate, and short-chain fatty acids (SCFAs) (Figs 4C and S3B). The biosynthetic and catabolic pathways of these metabolites are related to mitochondrial activity. Thus, these data further suggest that the metabolic phenotype in $Tbx1^{+/-}$ mice involves mitochondrial function.

RNA QuanTiseq analysis was performed on the contralateral brain hemispheres to those used for NMR (Group 3 in Fig S1C). We performed the sequencing on individual hemispheres; thus, each dataset was treated as a biological replicate. Results showed that vB12 treatment had a strong effect on both genotypes (Fig S4B and C). In contrast, in PBS-treated $Tbx1^{+/-}$ versus WT brains, we identified only 3 DEGs as detailed in the previous section (Fig S4D, Table S2).

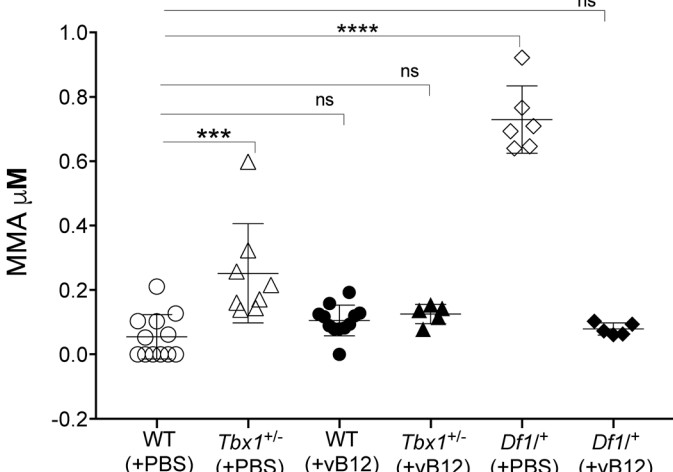

**Figure 3. Vitamin B12 treatment restores MMA to WT levels in *Tbx1*<sup>+/−</sup> and *Df1/*<sup>+</sup> mice.**
The abundance of MMA (μM, mean ± SEM) was evaluated in WT, *Tbx1*<sup>+/−</sup>, and *Df1/*<sup>+</sup> mice treated with vB12 or PBS (vehicle). Differences between groups were evaluated by the ordinary one-way ANOVA test and the Holm–Sidak's multiple comparison test (*$P < 0.05$, **$P < 0.01$, ***$P < 0.001$). The normal distribution was verified according to the D'Agostino and Pearson tests. Each symbol in the plot represents an individual animal.

In the brain of WT animals treated with vB12 versus PBS, we found 1831 DEGs (Figs 5A and S4C, Table S2), and in *Tbx1*<sup>+/−</sup> animals treated with vB12 versus PBS, we found 1,639 DEGs (Figs 5A and S4B, Table S2). Interestingly, vB12 treatment induced the altered expression of distinct groups of genes in WT and *Tbx1*<sup>+/−</sup> brains. A PCA of gene expression shows this clearly; that is, data points for PBS-treated *Tbx1*<sup>+/−</sup> and WT animals cluster together, as they do for vB12-treated *Tbx1*<sup>+/−</sup> and WT animals, but the respective clusters are distant from each other (Fig S4A). The response to vB12 treatment included three categories of DEGs: Group 1, common genes, that is, those that responded to treatment in both WT and *Tbx1*<sup>+/−</sup> samples (n = 816); Group 2, WT-unique genes, those that responded to treatment only in the WT samples (n = 1,015); and Group 3, *Tbx1*<sup>+/−</sup>-unique genes, those that responded to treatment exclusively in the *Tbx1*<sup>+/−</sup> samples (n = 823) (Fig 5A, Table S2).

To identify pathways that were altered by vB12 treatment, we performed a pathway analysis (using a DAVID functional annotation tool) of up-regulated and down-regulated genes separately for each of the three categories of DEGs. This revealed significant genotype-specific differences. Specifically, for Group 1 DEGs (common genes), the analysis identified pathways involved in Parkinson's disease, Huntington's disease, and oxidative phosphorylation (Fig 5B), which contained genes that were mostly down-regulated by vB12 treatment (Table S2). For Group 2 DEGs (WT-unique genes), the analysis did not identify any specific pathways. Conversely, for Group 3 DEGs (*Tbx1*<sup>+/−</sup>-unique genes), we identified pathways involved in the fatty acid metabolism (mainly down-regulated genes) and axon guidance process (up-regulated genes) (Fig 5B).

Collectively, these data suggest that *Tbx1* haploinsufficiency modifies the brain's transcriptional response to vB12 treatment. We speculate that in the *Tbx1*<sup>+/−</sup> brain, down-regulation of genes involved in fatty acid metabolism by vB12 reduces the accumulation of short-chain fatty acids and therefore corrects, in part, the metabolic imbalance observed in these mutants.

### PPI deficit in *Tbx1*<sup>+/−</sup> mice is rescued by vB12 treatment

The consequences of the metabolic imbalance observed in *Tbx1*<sup>+/−</sup> mice on brain health are unknown, but the presence of toxic metabolites such as MMA might affect brain function (25, 26). We therefore tested whether vB12 treatment was able to rescue the sensorimotor gating deficit (reduced PPI) that we reported previously in *Tbx1*<sup>+/−</sup> mice (2). The principle underlying the PPI paradigm is that an acoustic startle response, defined as a strong flinching response to an auditory stimulus, is attenuated when it is preceded by a weaker stimulus, the prepulse (Fig 6A, right panel). This attenuation is reduced in *Tbx1*<sup>+/−</sup> mice across a range of prepulse intensities (2). Reduced PPI is believed to be the result of deficits in the neuronal circuitry involved in sensorimotor gating, a pre-attentive brain function that facilitates the integration of motor and sensory stimuli. PPI was tested in *Tbx1*<sup>+/−</sup> and WT mice treated postnatally with vB12 or PBS. Specifically, we treated in parallel groups of *Tbx1*<sup>+/−</sup> mice with PBS (n = 14) or vB12 (n = 13) and compared them with a group of WT mice treated with PBS (n = 15) or vB12 (n = 18). Treatment was delivered intraperitoneally twice a week for 2 mo, starting at 4–5 wk of age, using the dosage of 20 mg/kg, as for the other experiments described here. The last injection of vB12 or PBS was administered one day before testing so that mice were off drug at the time of testing. Results showed that PBS-treated *Tbx1*<sup>+/−</sup> mice had impaired PPI, as expected (Figs 6B and S6), whereas in vB12-treated *Tbx1*<sup>+/−</sup> mice, the PPI response was indistinguishable from that of controls (Figs 6B and S7). No difference in PPI response was found in WT mice treated with vB12 compared with PBS-treated WT mice. Thus, prolonged postnatal vB12 treatment was able to rescue the PPI deficit in *Tbx1*<sup>+/−</sup> mice.

## Discussion

Our data show for the first time that *Tbx1* haploinsufficiency alters brain metabolism in mice. The most striking results obtained in this study were the accumulation of MMA in the brain of two mouse models of 22q11.2DS and the rescue of this anomaly by postnatal treatment with high doses of vB12. Thus, even though the two mutants are genetically and metabolically very different, in *Df1/*<sup>+</sup> mice, the MMA phenotype was not affected by heterozygosity of other genes in the deletion. MMA is produced by impaired catabolism of odd-chain fatty acids, cholesterol, valine, methionine, isoleucine, and threonine, with subsequent disruption of propionyl-CoA metabolism in the mitochondrial matrix and mitochondrial damage. Therefore, to have a broader picture of the brain metabolic profile in *Tbx1*<sup>+/−</sup> mice, we performed an unbiased metabolic analysis, as well as a candidate metabolite study. The results revealed reduced levels of leucine and increased levels of SCFAs. Catabolism of leucine produces acetyl-CoA, which acts as a primer for carbon chain elongation of fatty acids. Moreover, we observed that the treatment of *Tbx1*<sup>+/−</sup> mice with vB12 reduced MMA and restored SCFA concentrations to WT levels. Together, these data suggest that fatty acid synthesis and mitochondrial activity are

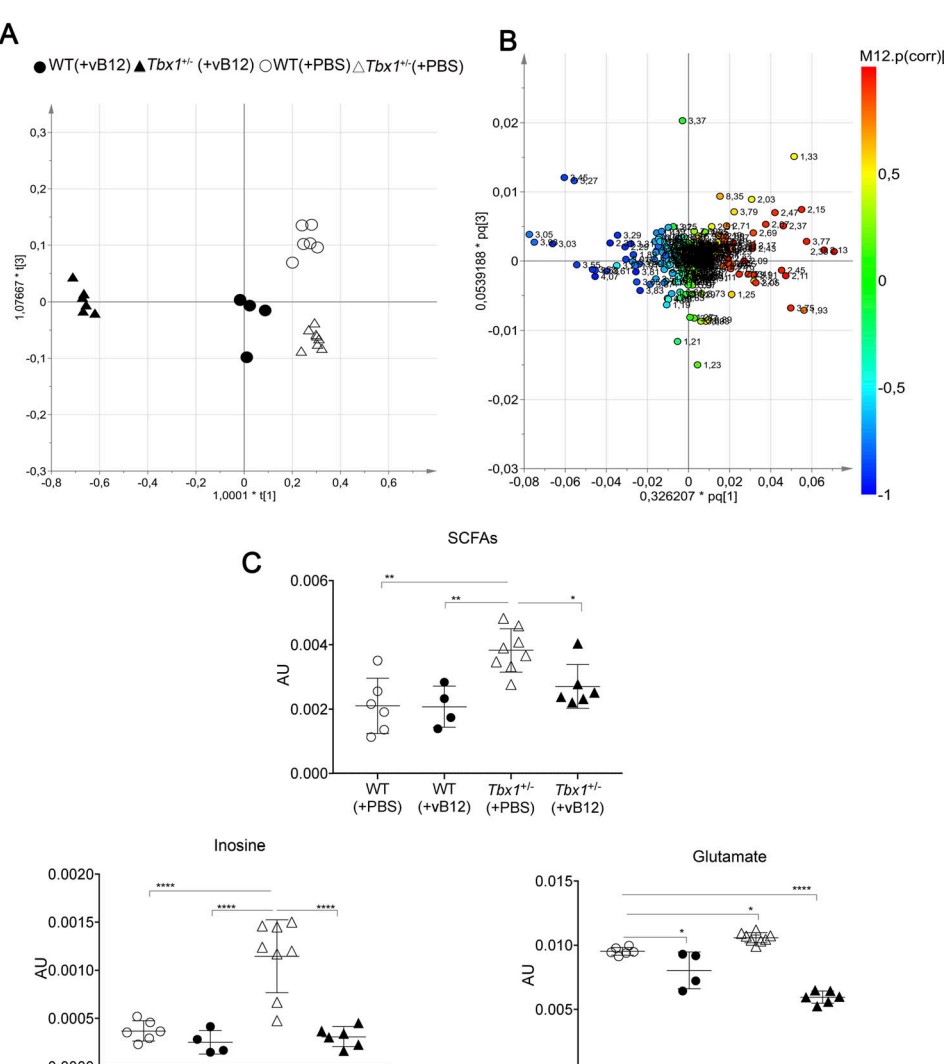

**Figure 4. Metabolic profile of the *Tbx1*<sup>+/−</sup> brain after vitamin B12 treatment.**

The metabolic profiles of *Tbx1* mutant brain extracts were compared to those of WT brains with and without vB12 treatment. **(A)** Score plot obtained from the orthogonal partial least squares discriminant analysis of brain extracts (NMR data). **(A, B)** Loading plot associated with the score plot in (A) showing the NMR signals responsible for the data cluster. Each variable in the plot corresponds to a signal in the metabolic profile of the dataset. **(C)** Graphical representation of metabolites that were rescued in *Tbx1*<sup>+/−</sup> brains after vB12 treatment (inosine, glutamate, and SCFAs). Normalized bin intensities corresponding to rescued molecules are expressed in arbitrary units (*$P$ < 0.01, **$P$ < 0.005, ***$P$ < 0.0005, ****$P$ < 0.0001).

impaired in the *Tbx1*$^{+/−}$ brain. Interestingly, TBX1 has already been linked to metabolism in adipose tissue, where its tissue-specific deletion in mice impaired the glycolytic pathway and fatty acid metabolism (27).

MMA is a toxic metabolite that impacts mitochondrial activity by impairing cell respiration and glutamate uptake (28). The consequence of the latter is to increase the concentration of extracellular glutamate (29). In the *Tbx1*$^{+/−}$ brain, in addition to increased MMA, we found several metabolites related to the glutamate pathway and to mitochondrial activity to be dysregulated. These included reduced levels of glutamine and adenosine and increased levels of glutamate and inosine. Glutamine is a necessary metabolite for the biosynthetic process involved in purine synthesis (adenosine and inosine) (30). Purine biosynthesis is also related to one-carbon metabolism, which generates different outputs that are required for nucleotide biosynthesis and for the maintenance of the redox (31). Moreover, purines play a role in neurotransmission and neurodevelopment (32). Therefore, the fact that vB12 treatment of

*Tbx1*$^{+/−}$ mice restored glutamate and inosine to WT levels is consistent with their metabolic relatedness and their dependence upon mitochondrial activity, which is responsive to vB12. vB12 supplementation has been shown to promote mitochondrial metabolism in human and mouse ileal epithelial cells (33).

Patients affected by methylmalonic aciduria show cognitive impairment, and in mice, MMA accumulation in the brain causes neuroinflammation and oxidative stress injury (34). An additional observation is that in normal physiological conditions, glutamate is maintained at low levels in the brain. It has been demonstrated that excess glutamate affects behaviour in mice (35). Indeed, massive release of glutamate can lead to excitotoxic brain damage in mice (36). Although both glutamine and glutamate were altered in *Tbx1*$^{+/−}$ mice, concentrations of the inhibitory neurotransmitter GABA only increased after vB12 treatment (Fig S3). Hence, after vB12 treatment, increased GABA, together with the restoration of normal (low) levels of glutamate, might counteract the potentially neurotoxic excitatory effects of increased glutamate in the *Tbx1*$^{+/−}$ brain.

# A

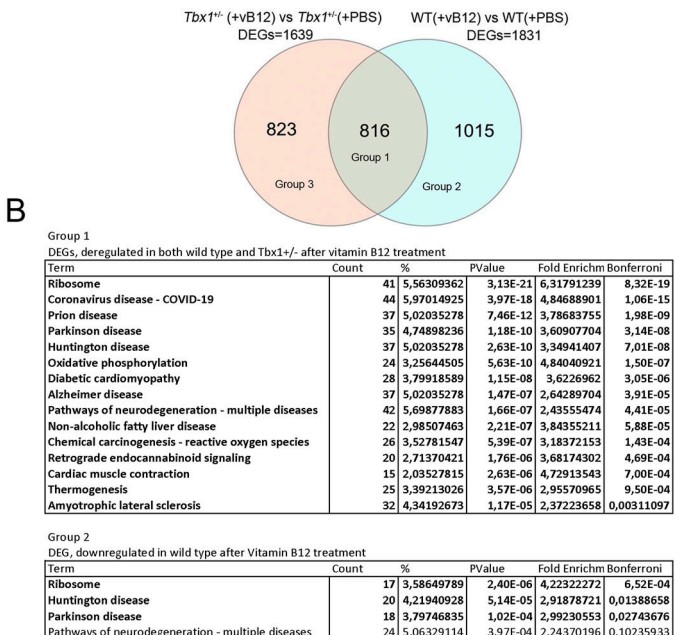

Venn diagram:
*Tbx1*+/- (+vB12) vs *Tbx1*+/- (+PBS) DEGs=1639
WT(+vB12) vs WT(+PBS) DEGs=1831

823 — Group 3
816 — Group 1
1015 — Group 2

# B

**Group 1**
DEGs, deregulated in both wild type and Tbx1+/- after vitamin B12 treatment

| Term | Count | % | PValue | Fold Enrich | Bonferroni |
|---|---|---|---|---|---|
| Ribosome | 41 | 5,56309362 | 3,13E-21 | 6,31791239 | 8,32E-19 |
| Coronavirus disease - COVID-19 | 44 | 5,97014925 | 3,97E-18 | 4,84688901 | 1,06E-15 |
| Prion disease | 37 | 5,02035278 | 7,46E-12 | 3,78683755 | 1,98E-09 |
| Parkinson disease | 35 | 4,74898236 | 1,18E-10 | 3,60907704 | 3,14E-08 |
| Huntington disease | 37 | 5,02035278 | 2,63E-10 | 3,34941407 | 7,01E-08 |
| Oxidative phosphorylation | 24 | 3,25644505 | 5,63E-10 | 4,84040921 | 1,50E-07 |
| Diabetic cardiomyopathy | 28 | 3,79918589 | 1,15E-08 | 3,6226962 | 3,05E-06 |
| Alzheimer disease | 37 | 5,02035278 | 1,47E-07 | 2,64289704 | 3,91E-05 |
| Pathways of neurodegeneration - multiple diseases | 42 | 5,69877883 | 1,66E-07 | 2,43555474 | 4,41E-05 |
| Non-alcoholic fatty liver disease | 22 | 2,98507463 | 2,21E-07 | 3,84355211 | 5,88E-05 |
| Chemical carcinogenesis - reactive oxygen species | 26 | 3,52781547 | 5,39E-07 | 3,18372153 | 1,43E-04 |
| Retrograde endocannabinoid signaling | 20 | 2,71370421 | 1,76E-06 | 3,68174302 | 4,69E-04 |
| Cardiac muscle contraction | 15 | 2,03527815 | 2,63E-06 | 4,72913543 | 7,00E-04 |
| Thermogenesis | 25 | 3,39213026 | 3,57E-06 | 2,95570965 | 9,50E-04 |
| Amyotrophic lateral sclerosis | 32 | 4,34192673 | 1,17E-05 | 2,37223658 | 0,00311097 |

**Group 2**
DEG, downregulated in wild type after Vitamin B12 treatment

| Term | Count | % | PValue | Fold Enrich | Bonferroni |
|---|---|---|---|---|---|
| Ribosome | 17 | 3,58649789 | 2,40E-06 | 4,22322272 | 6,52E-04 |
| Huntington disease | 20 | 4,21940928 | 5,14E-05 | 2,91878721 | 0,01388658 |
| Parkinson disease | 18 | 3,79746835 | 1,02E-04 | 2,99230553 | 0,02743676 |
| Pathways of neurodegeneration - multiple diseases | 24 | 5,06329114 | 3,97E-04 | 2,24370196 | 0,10235933 |
| Amyotrophic lateral sclerosis | 20 | 4,21940928 | 6,68E-04 | 2,39025006 | 0,16608617 |
| Non-alcoholic fatty liver disease | 12 | 2,53164557 | 8,20E-04 | 3,37984404 | 0,20005067 |
| Oxidative phosphorylation | 11 | 2,32067511 | 9,74E-04 | 3,57658741 | 0,23276953 |
| Prion disease | 16 | 3,37552743 | 0,00105021 | 2,63997768 | 0,24859348 |
| Coronavirus disease - COVID-19 | 15 | 3,16455696 | 0,00147518 | 2,6638329 | 0,33071495 |
| Alzheimer disease | 19 | 4,00843882 | 0,0025651 | 2,18795025 | 0,50272052 |

DEG, upregulated in wild type after Vitamin B12 treatment

| Term | Count | % | PValue | Fold Enrich | Bonferroni |
|---|---|---|---|---|---|
| Ras signaling pathway | 12 | 2,56410256 | 0,01115053 | 2,40406478 | 0,95670358 |
| Vascular smooth muscle contraction | 9 | 1,92307692 | 0,01127253 | 2,94247512 | 0,95817388 |
| Choline metabolism in cancer | 7 | 1,4957265 | 0,01745444 | 3,36282871 | 0,99277638 |
| Antifolate resistance | 4 | 0,85470085 | 0,02269202 | 6,49373821 | 0,99838265 |
| Yersinia infection | 8 | 1,70940171 | 0,02426459 | 2,78990234 | 0,99896966 |

**Group 3**
DEG, downregulated in *Tbx1*+/- after Vitamin B12 treatment

| Term | Count | % | PValue | Fold Enrich | Bonferroni |
|---|---|---|---|---|---|
| Fatty acid metabolism | 8 | 2,31884058 | 4,76E-05 | 8,19484737 | 0,01219492 |
| Parkinson disease | 12 | 3,47826087 | 0,00300011 | 2,86511581 | 0,53938498 |
| Butanoate metabolism | 4 | 1,15942029 | 0,00832876 | 9,40889883 | 0,88442317 |
| Biosynthesis of unsaturated fatty acids | 4 | 1,15942029 | 0,01573927 | 7,4717726 | 0,98331136 |

DEG, upregulated in *Tbx1*+/- after Vitamin B12 treatment

| Term | Count | % | PValue | Fold Enrich | Bonferroni |
|---|---|---|---|---|---|
| Axon guidance | 14 | 3,31753555 | 6,55E-05 | 3,8523408 | 0,01586708 |
| ATP-dependent chromatin remodeling | 9 | 2,13270142 | 0,00584191 | 3,29593653 | 0,76059686 |
| Long-term potentiation | 6 | 1,42180095 | 0,01071202 | 4,46017282 | 0,9277652 |

**Figure 5. Vitamin B12 treatment has a genotype-specific impact on the brain transcriptional profile.**
**(A)** Venn diagram representing the intersection of two groups of differentially expressed genes (DEGs): *Tbx1*+/−(+vB12) versus *Tbx1*+/− (+PBS) and WT(+vB12) versus WT(+PBS). Groups 1, 2, and 3 are defined in (B). **(B)** Gene ontology analysis (DAVID) of differentially down-regulated and up-regulated genes in each of the indicated groups: Group 1—differentially expressed genes deregulated in both WT and *Tbx1*+/− after vB12 treatment; Group 2—differentially expressed genes deregulated exclusively in WT after treatment; and Group 3—differentially expressed genes deregulated exclusively in *Tbx1*+/− after vB12 treatment.

Another important finding of our study was that postnatal vB12 treatment of *Tbx1*+/− mice rescued the PPI deficit previously reported (2). The mechanisms by which *Tbx1* mutation affects brain development and brain function are unknown, but they are non–cell-autonomous mechanisms (37) because in the mouse brain, *Tbx1* expression is limited to brain vessels, and it is not expressed in neurons or other brain cell types (2, 38) (https:// portals.broadinstitute.org/single_cell/study/aging-mouse-brain). Moreover, not all brain ECs express *Tbx1*. Single-cell transcriptomic analysis of the adult mouse brain indicates that although ECs comprise ~6% of brain cells (24), only 1.5% expresses *Tbx1*, all of which are ECs. Despite this low level of expression, reduced *Tbx1* dosage in adult heterozygous mice leads to mild brain vessel hyperbranching and disorganization of the entire brain vessel network, as well as reduced brain vessel perfusion, and tissue hypoxia (38). Gross disruption of the blood–brain barrier was not found. Nevertheless, changes in the endothelial compartment caused by *Tbx1* mutation could affect brain metabolism by altering signals or metabolite exchange across cell membranes, which could in turn cause or contribute to the diffuse brain phenotypes observed in *Tbx1*+/− mice. In fact, we think that the phenotypic rescue might occur through a metabolic route rather than a vascular route, because vB12 treatment began in juvenile mice, when the brain vascular network is fully formed. Despite these considerations, our study does not establish a causal relationship between elevated brain MMA and PPI impairment, nor does it confirm that the rescue of PPI impairment by vitamin B12 is due to reduced MMA levels.

Could the behavioural phenotype in *Tbx1*+/− mice be due to altered levels of glutamate and glutamine reported here? A recent study reported that 22q11.2DS patients with psychotic symptoms had increased levels of glutamate and glutamine in the hippocampus and in the superior temporal cortex (39). Previously, increased glutamate and myo-inositol were found in the hippocampus of 22q11.2DS patients with schizophrenia (40). Future studies will be needed to elucidate the role of these metabolites in the behavioural anomalies in the 22q11.2DS mouse models.

In summary, our study shows for the first time that in adult mice, *Tbx1* haploinsufficiency alters brain metabolic homeostasis and that this is partially rescued by postnatal vB12 treatment. In addition, we report that the same pharmacological treatment rescues PPI deficits in *Tbx1* mutants, suggesting that they are related to the underlying metabolic changes caused by the mutation. The definition of the molecular mechanisms that lead to these metabolic anomalies will require further studies. Nevertheless, our data open a new direction for the design of a therapeutic strategy that could be applied postnatally and might help to combat the brain-related phenotype of 22q11.2DS.

# Materials and Methods

## Mouse lines

Animal research was conducted according to EU and Italian regulations. The animal protocol has been approved by the animal welfare committee of the Institute of Genetics and Biophysics (Organismo per il Benessere Animale, OPBA-IGB), protocol n. 7E58D.24. We used the following mouse lines in a C57BL/6N background: *Tbx1*+/− (*Tbx1* lacZ/+ or *Tbx1*ΔE5/+), *Df1*/+ (41, 42). Group 2 mice were obtained by intercrossing *Tbx1*+/− and *Df1*/+ mice.

## Mouse treatment

For all experimental procedures, *Tbx1*+/− and WT mice were injected intraperitoneally with vB12 dissolved in PBS 20 mg/kg (stored away from light) or with PBS, twice a week for 4 wk starting at 4 wk of age.

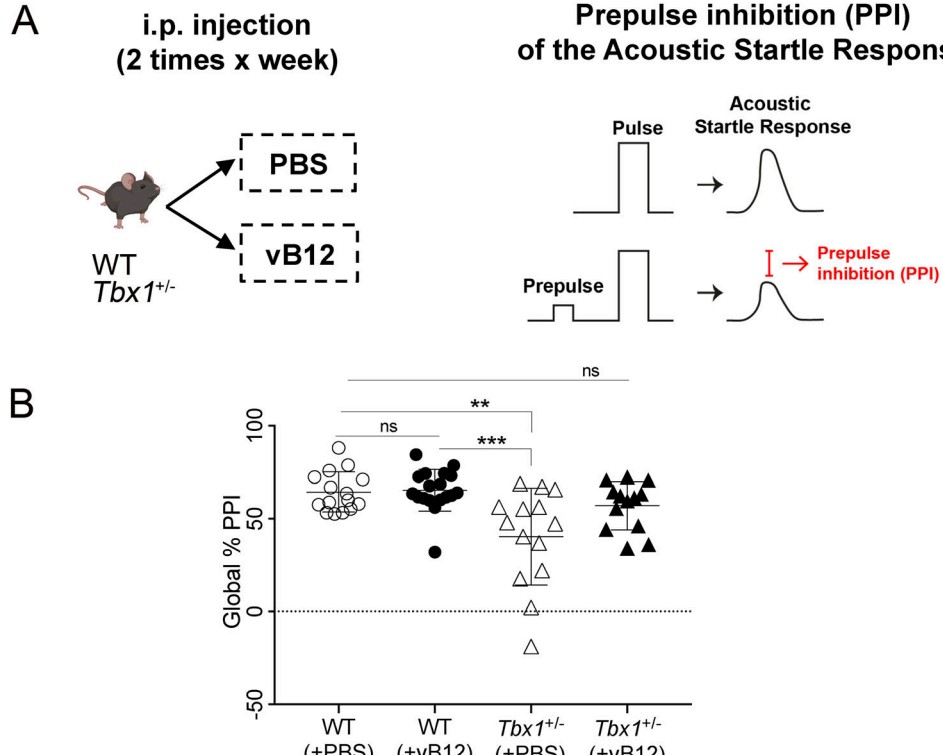

**A**

**i.p. injection**
**(2 times x week)**

WT
*Tbx1*⁺/⁻

PBS

vB12

**Prepulse inhibition (PPI)**
**of the Acoustic Startle Response**

Acoustic
Startle Response

Pulse

Prepulse

→

→

Prepulse
inhibition (PPI)

**B**

Global % PPI

WT
(+PBS)

WT
(+vB12)

*Tbx1*⁺/⁻
(+PBS)

*Tbx1*⁺/⁻
(+vB12)

**Figure 6. Postnatal vitamin B12 treatment ameliorates the PPI impairment in *Tbx1*⁺/⁻ mice.**
**(A)** Schematic representation of the experimental procedure. **(B)** Dot plots showing global % PPI in WT (n = 15) and *Tbx1*⁺/⁻ (n = 14) mice treated with PBS, and in WT (n = 18) and *Tbx1*⁺/⁻ mice (n = 13) treated with vB12. $**P < 0.005$, $***P < 0.0005$ (Bonferroni's post hoc test).

For behavioural analysis, animals were treated until the day before behavioural testing that was performed at 12–13 wk of age. For metabolomic and molecular studies, at the end of treatment (at 8 wk of age), the brains were surgically removed after cervical dislocation and stored at –80°C.

## Metabolomic analysis of mouse brains

### LC-MS/MS

Metabolites were extracted from mouse brain tissues (*Tbx1*⁺/⁻ and WT) for targeted metabolome analysis by liquid chromatography–tandem mass spectrometry (LC-MS/MS). Frozen tissues were disrupted in 50:50 cold methanol/0.1 M hydrogen chloride and homogenized at high speed by shaking with stainless-steel beads into a TissueLyser LT (QIAGEN). The metabolite-containing supernatant was isolated from the protein pellet by centrifugation at 16,006*g* for 60 min at 4°C. Proteins were resuspended and quantified for the subsequent normalization data as previously published (43). The metabolite extracts were alkalinized to pH 7–8 and dried under nitrogen. Finally, dried metabolite mixtures were resuspended in methanol and analysed by targeted MS/MS to determine the contents of amino acids (AAs), acylcarnitines (ACs), MMA, and lactic acid. The metabolomic platform, which was developed to enable the rapid identification of biomarkers of inherited metabolic diseases, was adapted for the purpose of the present study. Before MS analysis, a standard mixture containing labelled AAs and ACs was added to all of the samples for the derivatization of all the endogenous and exogenous molecules, as previously reported (44). Finally, the MS/MS

experiments were carried out in flow injection analysis mode for AAs and ACs and in LC for MMA and lactic acid (LA) using an API 4000 triple quadrupole mass spectrometer (Sciex) coupled with a 1160 series Agilent HPLC system (Agilent Technologies) (45, 46).

The chromatographic separation of MMA and LA was performed with a 3-µm analytical column Phenomenex Gemini C6-Phenyl 100 × 2.0 mm (Phenomenex). Solvents A and B consisted of 0.1% formic acid in water and acetonitrile, respectively. The flow was set at 300 µl/min with the following gradient: 0.0–0.5 min: 30% B; 0.5–3.0 min: 30% to 90% B; 3.0–3.10 min: 90% to 30% B; and 3.1–4.0 min: 30% B. Multiple reaction monitoring defined experimental parameters were as follows: polarity: negative; Q1/Q3 (m/z): 117.00/73.00 (MMA), 89.00/59.00 (LA); DP (volts): −34 (MMA), −50 (LA); and CE (volts): −12 (MMA), −14 (LA). Quantitative analysis of the data was performed with ChemoView v2.0.4 software through the comparison of the analytes and their corresponding internal standard areas.

## Multivariate statistical data analysis of LC-MS/MS data

Univariate statistical data analysis was carried out with GraphPad Prism 9.0, and the results are presented as the mean ± SEM. The data were normalized to convert values from different datasets to a common scale, defining zero as the smallest value and 100 as the largest value in each dataset. The normalized metabolic dataset was $\log_{10}$-transformed. The volcano plot was performed to visualize the predictive component loading and identify significantly altered metabolites by their content variation (Difference) and statistical significance (–$\log_{10}$ P-value). The normalized dataset was then used to perform pattern correlation analyses with MetaboAnalyst 5.0

([http://www.metaboanalyst.ca](http://www.metaboanalyst.ca)). The statistical significance of the difference in metabolite sample concentrations between two different groups was evaluated by parametric (unpaired *t* test with Welch's correction) or non-parametric (Mann–Whitney test) comparisons. The significant difference between multiple groups was evaluated by ordinary one-way ANOVA and Holm–Sidak's multiple comparison test in normally distributed datasets, and Kruskal–Wallis test and Dunn's multiple comparison test in non-normally distributed datasets. The normal distribution was verified according to the D'Agostino and Pearson test.

### NMR

Combined extraction of polar and lipophilic metabolites was carried out using methanol/water/chloroform as suggested by the Standard Metabolic Reporting Structures working group (47). Polar and nonpolar fractions were then transferred into glass vials, and the solvents were removed using a rotary vacuum evaporator at room temperature and stored at –80°C until they were analysed. For NMR analysis, polar fractions were resuspended in 630 μl of PBS (pH 7.4), adding 70 μl of $^2H_2O$ solution (containing 1 mM sodium 3-trimethylsilyl [2,2,3,3-2H4]propionate (TSP) as a chemical shift reference for $^1H$ spectra) to provide a field frequency lock, reaching 700 μl of total volume. The nonpolar fractions were resuspended in 700 μl of $C^2HCl_3$. Samples were loaded into the autosampler and NMR spectra and acquired on a Bruker Avance III 600-MHz spectrometer (Bruker BioSpin GmbH), equipped with a TCI CryoProbe fitted with a gradient along the z-axis, at a probe temperature of 27°C. In particular, standard 1D proton spectra and 2D experiments (clean total-correlation spectroscopy (TOCSY) and heteronuclear single quantum coherence (HSQC)) were acquired providing one-dimensional metabolic profiles and homonuclear and heteronuclear spectra for metabolite identification. Metabolite assignments were achieved by comparing signal chemical shifts with published data and online databases. All 1D spectra were processed, and automatically, data were reduced in bins, arranged as a data matrix with the AMIX 3.9.7 package (Bruker BioSpin GmbH), and then imported into the SIMCA14 package (Umetrics) for multivariate data analysis.

### Multivariate statistical data analysis of NMR data

A multivariate statistical analysis was performed to discriminate between PBS- and vB12-treated mice. In particular, the unsupervised PCA was first applied to assess class homogeneity, uncover data trends, and detect outliers. Then, supervised methods such as OPLS-DA were used to improve class separation, thus better appreciating clusters and the spectral variables influencing sample distribution according to the alteration of their metabolic profile. The performance of each supervised model was estimated by evaluating the per cent of data variation explained ($R^2$) and the one predicted by according to cross-validation (Q2). Moreover, OPLS-DA models were validated by internal iterative cross-validation with seven rounds of permutation test response (800 repeats) not clear, and CV-ANOVA (ANOVA testing of cross-validated predictive residuals). Data visualization was achieved through score, loading,

and S plots, highlighting specific compounds as putative markers. To identify a subset of the most responsible metabolites characterizing class discrimination, NMR variables were selected considering the correlation loading values |p(corr)|> 0.7 (as shown in the S plots in the supplementary figures), and we tested them for univariate statistical analysis using an unpaired *t* test for two-class analysis (once considered variance homogeneity and Welch correction) or ANOVA test with a Bonferroni correction in case of multiple comparisons, after assessing Gaussian distribution with a normality test (Shapiro–Wilk test). Not clear statistical tests were elaborated on with the OriginPro 9.1 software package (OriginLab Corporation) and R software (R Core Team; [https://www.r-project.org/](https://www.r-project.org/)).

### Pathway analysis

To identify the most relevant pathways involved in the NMR study, the more significant metabolites in the class separation were analysed by MetaboAnalyst 5.0 (48).

The pathway analysis approach integrates results from both pathway enrichment analysis (KEGG) and pathway topology analysis to identify the most relevant pathways involved and was analysed using Fisher's exact test for overrepresentation and relative betweenness centrality for pathway topology analysis.

### RNA isolation

Total RNA was isolated from whole brains with QIAzol (QIAGEN) according to the manufacturer's protocol and with an additional precipitation step.

### Transcriptomic QuanTiseq analysis

QuanTiseq libraries were prepared and sequenced at the Telethon Institute of Genetics and Medicine (TIGEM), Naples. The facility performed quality control and trimming for each sample, aligned them to the mouse genome (mm10) using STAR (49), and provided the gene expression data as a raw count matrix. We imported the count matrix in the R environment, evaluated the counts per million expression, and retained in our subsequent analysis only those genes with a counts per million > 1 (as expressed genes) in at least three samples. We used the edgeR package (50); that is, we first estimated the library size using the calcNormFactors function. Then, we defined the design matrix, with the three experimental conditions and the corresponding biological replicates, estimating the parameters of the negative binomial model using the estimateGLMRobustDisp function. Finally, we fitted the model with the glmFit function and extracted the contrast from the joint fit. We used the glmLRT function to test the differential expression of genes, and we applied the decideTests function with method = "separate" and FDR < 0.05 for each contrast. Pathway analysis was performed using the gProfiler online tool. In all conditions, the lists of DEGs were analysed, using all expressed genes as the background.

## Behavioural analysis

### PPI–acoustic startle response and PPI

The test was performed using the SR-LAB Startle Response System apparatus (San Diego Instruments), as previously described (51, 52). Briefly, the PPI session started with a 5-min habituation period, in which only a background noise (BN) of 72–73 dB was presented, followed by 5 consecutive startle trials, which were excluded from the statistical analysis, and 9 different trial types were repeated 10 times in random order and with an intertrial interval of 25 s on average. The startle trial consisted of the presentation of a 110-dB/40-ms acoustic pulse (white noise); the "no-stimulus" trial consisted of BN/10-ms presentation to measure the baseline movement of the mouse; the 4 prepulse-only trials consisted of the presentation of 10-ms-long 76-, 80-, 88-, or 100-dB stimuli; the 3 PPI trials consisted of 10-ms-long stimuli of 76, 80, or 88 dB followed after 50 ms by a 110-dB/40-ms pulse. For each trial, the startle response was recorded every millisecond for 65 ms after the onset of the acoustic pulse. Maximal peak-to-peak amplitude was used to determine the ASR in the acoustic startle pulse and/or prepulse-alone trials. Mice were off drug during the whole procedure.

PPI was expressed as %PPI = (100 − [startle response for prepulse/startle response for startle-alone trials] × 100). The whole PPI session lasted about 45 min. Average % PPI (global % PPI) over the sampling period was taken as the dependent variable (53).

Percentage PPI was analysed with a three-way ANOVA for repeated measures for the factors: genotype (two levels: WT and $Tbx1+/−$) and treatment (PBS, vB12); and prepulse (repeated measures: % PPI 76p110, % PPI 80p110, % PPI 88p110) and treatment (three levels: WT PBS, $Tbx1^{+/−}$ PBS, and $Tbx1^{+/−}$ vB12). A Bonferroni correction was applied for global % PPI.

## Supplementary Information

## Acknowledgements

We thank Rosa Ferrentino and Marchesa Bilio for expert laboratory assistance. We also thank IGB mouse facility and especially Lucia Mele for expert technical assistance. This work was funded by the Ministry of University and Research (MUR), National Recovery and Resilience Plan (NRRP) 2022, project MNESYS (PE0000006) to A Baldini; PRIN-PNRR 2022, project P2022ARZ5J to G Lania, M Ruoppolo, and E Illingworth; and DBA.AD005.225 NutrAge FOE 2021 to E De Leonibus.

## Author Contributions

M Caterino: formal analysis and investigation.
D Paris: formal analysis, investigation, methodology, and writing—review and editing.
G Torromino: formal analysis, validation, investigation, and writing—review and editing.
M Costanzo: investigation.
G Flore: resources.
A Tramice: resources.
E Golini: investigation.
S Mandillo: investigation.
D Cavezza: investigation.
C Angelini: data curation, formal analysis, and writing—review and editing.
M Ruoppolo: resources, supervision, validation, and writing—review and editing.
A Motta: resources, supervision, validation, and writing—review and editing.
E De Leonibus: resources, supervision, validation, and writing—review and editing.
A Baldini: resources, funding acquisition, and writing—review and editing.
E Illingworth: conceptualization, funding acquisition, and writing—original draft, review, and editing.
G Lania: conceptualization, supervision, funding acquisition, investigation, visualization, project administration, and writing—original draft, review, and editing.

## Conflict of Interest Statement

The authors declare that they have no conflict of interest.

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
