## [Reviewer comments · Life Science Alliance]

Life Science Alliance

Brain and behavior anomalies caused by Tbx1 haploinsufficiency are corrected by vitamin B12

Marianna Caterino, Debora Paris, giulia torromino, Michele Costanzo, Gemma Flore, Annabella Tramice, Elisabetta Golini, Silvia Mandillo, Diletta Cavezza, Claudia Angelini, Margherita Ruoppolo, Andrea Motta, Elvira De Leonibus, Antonio Baldini, Elizabeth Illingworth, and Gabriella Lania

DOI: <https://doi.org/10.26508/lsa.202403075>

Corresponding author(s): Gabriella Lania, Institute of Genetics and Biophysics and Elizabeth Illingworth, university of salerno

Review Timeline:	Submission Date:	2024-10-07
	Editorial Decision:	2024-10-23
	Revision Received:	2024-11-07
	Accepted:	2024-11-07

Transaction Report:

Please note that the manuscript was reviewed at Review Commons and these reports were taken into account in the decision-making process at Life Science Alliance.

Review
COMMONS

Revision Plan

Manuscript number: RC-2024-02422

Corresponding author(s): First name: Marianna Caterino, Last name, Gabriella Lania

[The “revision plan” should delineate the revisions that authors intend to carry out in response to the points raised by the referees. It also provides the authors with the opportunity to explain their view of the paper and of the referee reports.]

The document is important for the editors of affiliate journals when they make a first decision on the transferred manuscript. It will also be useful to readers of the reprint and help them to obtain a balanced view of the paper.

1. General Statements [optional]

This section is optional. Insert here any general statements you wish to make about the goal of the study or about the reviews.

The goal of the study was to obtain the brain metabolic profile of Tbx1 heterozygous mice (adult) and determine how this is altered by postnatal vitamin B12 treatment. The study uses a combination of mass spectrometry and nuclear magnetic resonance for the metabolic profiling of whole brain tissue. We also performed a transcriptomic analysis of a portion of the brain tissue samples using RNA seq (bulk). The major findings of the study are that i) Tbx1^{+/-} mice have increased levels of a brain toxic metabolite, methylmalonic acid, which are normalized by vitamin B12 treatment, ii) they also have a more generalized brain metabolic phenotype which is partially responsive to vitamin B12 treatment, iii) transcriptomic evidence of a genotype – vB12 treatment interaction.

The reviewers (three) all appreciated the importance and interest of the study, while highlighting shortcomings in the experimental planning and data interpretation. The major issue was the absence of what was considered by two of the reviewers to be an essential control group, i.e., wild type mice treated with vitamin B12. This group was not included in the data analysis in the original manuscript. In the revised manuscript this group has been added and all of the relevant experimental data reanalyzed.

The inclusion of this control group has improved the quality and clarity of the manuscript. Because vitamin B12 has pleiotropic effects on metabolism and gene transcription that are unrelated to Tbx1 mutation, the inclusion of this control group has, inevitably, exerted a “dilution effect” on some of the data. Nevertheless, the overall message and conclusions of the study are unchanged.

2. Description of the planned revisions

Insert here a point-by-point reply that explains what revisions, additional experimentations and analyses are planned to address the points raised by the referees.

In the revised manuscript, we have responded to the Reviewers comments in a point-by-point reply, which is detailed in section 3.

There are no additional revisions planned at this point.

3. Description of the revisions that have already been incorporated in the transferred manuscript

Please insert a point-by-point reply describing the revisions that were already carried out and included in the transferred manuscript. If no revisions have been carried out yet, please leave this section empty.

Response to Reviewer #1

Major comments:

1. The connection between vB12 and MMA is weak, and the attempt to connect these pieces to PPI seems somehow forced. For instance, the authors do not convincingly demonstrate that MMA causes the PPI deficit. Furthermore, vB12

Revision Plan

may rescue PPI independently of MMA. The authors should be more transparent about the lack of connection or causality between changes in metabolism and behavior.

We appreciate the reviewer's comment and acknowledge that we have not demonstrated causal relationships between increased MMA, PPI deficits in *Tbx1*^{+/-} mice and their rescue by vB12. They are associations.

In the revised manuscript, we have clarified this in the Discussion, para.4, by adding the following phrase. "The results of our study do not prove a causal relationship between elevated brain MMA and PPI impairment, nor do they tell us whether rescue of the PPI impairment by vB12 occurs by reducing MMA".

Regarding the comment of a weak connection between vitamin B12 and MMA, we respectfully disagree.

The biochemistry underlying the connection is outlined clearly in the Introduction, page 4, para. 2.

Patients with vitamin B12 deficiency typically exhibit elevated levels of MMA and administration of vitamin B12 (cobalamin) helps to normalize MMA values (Robert & Brown, 2003). Furthermore, several animal models with genetic alterations in the vitamin B12 pathway exhibit high levels of MMA. For instance, mice lacking the cobalamin transporter have increased MMA (Bernard et al., 2018). Additionally, mice lacking the mutase (Mut), which requires vitamin B12 as a cofactor for the conversion of methylmalonyl CoA to succinyl-CoA for entry into the Krebs cycle, demonstrate elevated levels of MMA and are unresponsive to vitamin B12 treatment (Peters et al. 2006). In the revised manuscript, we have cited these references (Introduction, page 4, para. 2).

2. Throughout the manuscript, an important control is missing: WT+ vB12 group. Data from this group should be added to Figures 1, 3, 4, 6, and Supplemental Figures 3 and 4 to show the effect of vB12 on WT mice.

All of the experiments reported in the original manuscript included this control group although it was not always included in the data analysis and therefore in the figures, as observed by the reviewer. In the revised manuscript all of the relevant figures and tables now include these data.

3. In Figures 1C and 3, data from the respective WT controls in the Df1 and Tbx1 cohorts should be shown.

The wild type (WT) animals serve as the control group for both *Tbx1*^{+/-} and *Df1*^{+/+} mice because they were littermates, obtained from matings between *Df1*^{+/+} and *Tbx1*^{+/-} mice. This has been clarified in the Materials and Methods, and in a new cartoon which has been added to Supplementary Figure 1 (1C) showing all of the animal groups used for the various studies (NMR, transcriptomics, behavior).

4. The Supplemental Table S1 includes 17 WT controls for Tbx1+/- and 6 WT controls for Df1/+, but Figure 1C includes only one group of 11 WT controls. For which group were those 11 WT controls?

5. Here are several examples of inconsistency in the data: "For this, we first performed a preliminary metabolome analysis using isolated whole brains of male and female Tbx1+/- (n= 18) and WT (n= 10) mice between one and two months of age. A set of metabolites was quantified in brain extracts by liquid chromatography tandem mass spectrometry (LC-MS/MS) (Supplementary Table 1)." Again, the number of mice the authors note in the manuscript does not match that shown in Supplementary Table 1 (Tbx1+/- (n=14) and WT (N=17)). "We analyzed whole brain tissue isolated from Df1/+/+ and WT (control) mice (n = 5 per genotype)." Again, the numbers of mice do not match those in Supplementary Table 1, which notes 6 Df1/+/+ mice and 6 WT mice.

We apologize for these errors and inconsistencies in the text and tables, all of which have been corrected in the revised manuscript. In addition, we have added the aforementioned cartoon (Supplementary Figure 1C) and we have improved the presentation of the data (genotypes and treatments) in Supplementary Tables 1 and 2. We hope that these changes provide the expected clarity to the data.

6. MMA is the only metabolite that is similarly changed between Tbx1+/- and Df1/+ brains. This is an interesting observation. However, there is no other overlap in metabolic changes between these two mutants. This is a concern that requires clarification

We appreciate the reviewer's comment. The observation is not altogether surprising considering that the *Df1* deletion includes at least 9 genes involved in metabolic pathways (cited refs (Maynard, 2008; Meechan, 2011; Devaraju and Zakharenko, 2017)) any of which might counteract or compensate for changes caused by *Tbx1* mutation alone. In addition, heterozygosity for other genes in the deleted region (*Df1* encompasses over 20 genes) might affect metabolic processes indirectly. In the revised manuscript we have added the following phrase to the Discussion, para.1 "Thus, even though the two mutants are genetically and metabolically very different, in *Df1*^{+/+} mice, the MMA phenotype is not affected by heterozygosity of other genes in the deletion.

7. The authors mention that MMA is not changed in pre-term Tbx1+/- embryos, but no data are provided. What about MMA levels in Df1/+ embryos?

Revision Plan

In pre-term $Tbx1^{+/-}$ and WT embryos MMA was undetectable. This is now stated in the final paragraph of the first section of the Results.

We did not measure MMA in $Df1/+$ embryos. It was not a goal of the study to compare the metabolome of these two genetically very different mutants. The MMA data on $Df1/+$ mice are presented because they show the potential relevance of this phenotype for the human disease, and they justify the use of the single gene ($Tbx1^{+/-}$) mutants for studies into metabolism-related disease mechanisms. See also response to point 12

8. *In some cases, the differences in metabolites (e.g., glutamine, glutamate, phosphoethanolamine, taurine, leucine, myo-inositol, and niacinamide) between the WT and $Tbx1^{+/-}$ mice is very minimal (Supplemental Figure 2). The y-axis scale should start at 0.*

We have changed the y-axis settings where necessary

9. *The vB12 is administered via two different regimens: 1) every 3 days for 28 days, at 4–8 weeks of age for metabolic measurements, and 2) twice a week for 2 months for PPI behavioral testing. Is there any reason the authors chose different protocols?*

We apologize for the confusion, which was due to an oversight in the Materials and Methods section that has been corrected. The weekly injection regimen was the same for mice used in the behavioral and metabolic studies, but the treatment time was shorter for the metabolic studies, for practical reasons beyond our control; mice received vB12 injections twice a week, beginning at 4 wks of age and continuing until 8 wks or 12 wks of age for metabolic and behavioral studies respectively.

10. *The authors should add the following references to the study: Long et al., Neurogenetics (2006), which shows no change in PPI in $Tbx1^{+/-}$ mice. This discrepancy compared with the current study results and those of Paylor et al., Proc Natl Acad Sci U S A (2006) should be discussed.*

We have not cited the study by Long et al. because there are no obvious reasons for the discrepancy (age, mouse strain, sex) that could be discussed. Beyond this of course we cannot comment on data generated by another research group. Nevertheless, the presence of the PPI deficit in $Tbx1^{+/-}$ mice has been confirmed in two different $Tbx1$ alleles $Tbx1^{lacZ/+}$ and $Tbx1^{\Delta E5/+}$, by two different investigators, Dr. Richard Paylor using $Tbx1^{lacZ/+}$ mice (Paylor et al. 2001) and Dr. Elvira De Leonibus (co-author of this manuscript) using $Tbx1^{\Delta E5/+}$ mice, in two different countries (USA and Italy) in a rederived colony of mice.

11. *Figure 6B is a concern. The PPI decrease in the $Tbx1^{+/-}$ group appears to be driven by results from 3–4 mice. First, are those data statistical outliers?*

With all due respect, this is not the case. Eight $Tbx1^{+/-}$ mice, i.e., >50% of those tested had PPI values below the minimum observed in WT mice. The behavioural data were checked for the presence of outliers in each group using the Grubbs test, which yielded negative results. Our finding of PPI deficits in $Tbx1^{+/-}$ mice are in line with previously published data in $Tbx1^{+/-}$ and other animal models of 22q11.2 microdeletion (Paylor et al., 2006; Paylor and Lindsay, 2006; Stark et al., 2008), as well as in humans (Sobin et al., 2005).

Second, experiments in the same mice would be more informative. Do PBS-treated mutants recover PPI if they are treated with vB12 and vice versa? If the authors are concerned about the age difference, they also should include age-dependent effects on PPI.

We decline to perform the proposed experiment for the reasons described in section 4 of the Revision Plan

12. *Because vB12 treatment completely rescued the MMA level in $Df1/+$ mice (Figure 3), the authors should include a figure showing PPI test results in $Df1/+$ mice.*

Vitamin B12 treatment fully rescued the MMA phenotype in both mutants (Figure 3). Whether it rescues the PPI defect in $Df1/+$ mice is not important for this study. We used $Df1/+$ mice as an entry point, in order to give validity to the pursuit of the MMA phenotype in the single gene mutant ($Tbx1^{+/-}$), in which we expect that it will be easier to find disease mechanisms. For this reason, we focused our attention on identifying metabolic alterations in adult $Tbx1^{+/-}$ mice.

See also response to point 7.

13. *Figure 1A and B table: Did the authors mean Log2FC instead of FC? The authors also should present the source data by adding supplemental tables that include raw data and normalized conversion, etc., as described in the multivariate statistical data analysis of the LC-MS/MS data.*

The new Figures 1A, Figure 3 and the accompanying tables now state Log2FC. New Supplementary Table 1 presents the raw data that were normalized on the basis of the amount of protein in the samples, described and referenced in the Materials and Methods

Revision Plan

14. "...we identified a new metabolic phenotype that was associated with reduced sensorimotor gating deficits in *Tbx1*^{+/-} mice". Although the authors showed the PPI rescue by treating *Tbx1*^{+/-} mice with vB12, that result alone does not prove the association of metabolic phenotype with sensorimotor-gating deficit; other supporting data are needed.

This is perhaps a question of semantics; by associated we mean that the two phenotypes, metabolic alterations and reduced PPI were observed together

15. The authors stated, "Results showed that there were very few differentially expressed genes in *Tbx1*^{+/-} vs WT brains, (n=22 out of 14535 expressed genes (Fig. 5 and Supplementary Tab. 2)". However, they described how 3 differentially expressed genes are involved in mitochondrial activity in the Discussion. The authors should describe those 3 genes and their relation to the metabolic change.

The results that the reviewer refers to have changed in the revised manuscript due to the inclusion of the control group WT +vB12 in the data analysis. The transcriptome analysis revealed that vB12 had a stronger impact than genotype, and as a consequence, the statistical analysis of all groups did not highlight minor differences between the two genotypes.

16. Figure 5B: The authors claimed that they detected similar transcription profiles between WT+vB12 vs. *Tbx1*^{+/-}+vB12, comparing *Tbx1*^{+/-}+PBS vs. *Tbx1*^{+/-}+vB12. This is based on 947 genes being downregulated and 834 being upregulated, which is not appropriate. The authors should normalize those data to the numbers of genes upregulated and downregulated in WT+PBS vs. WT+vB12 respective groups.

We said that we detected similar transcription profiles in PBS-treated WT and *Tbx1*^{+/-} brains; a WT+vB12 group was not present. The latter group is included in revised manuscript and the data reanalyzed comparing all groups. See also response to points 2 and 15.

Minor comments

1. Supplementary Table S1 shows the identical MMA concentration "0.2" for 6 controls. Is this correct?
This was an error that has been corrected; the value is 0.00 (not detectable).

2. Remove the callout for Figure 1C at the end of the second paragraph in Results.
This figure is no longer present

3. There are multiple typos throughout the manuscript.
Here are several examples:

a. Fig1B graph- Df/+ => Df1/+
Figure changed in revised manuscript

b. "Together, the hydrophilic and lipophilic results revealed a group of 6 compounds that characterized the brain metabolic differences between *Tbx1*^{+/-} and WT mice (Figure 2B, 2C)". Figure 2A should be included also.
Corrected

c. "In support of this notion, is the finding that...(remove)
Removed

d. Remove double periods: "The pathways found are depicted in Figure 2C' which reports the impact of each pathway versus p values.."
Corrected

e. Panel labels in all figures are misplaced.
Panel labels are aligned correctly

We have performed a spelling and grammar check on the text

f. "In support of this notion... at least nine orthologs are involved in mitochondrial metabolism". What are those 9 mitochondrial genes? Kolar et al., *Schizophr Bull* (2023) indicates that there are 8 mitochondrial genes within the 22q11.2 locus. The authors need to list these genes.

This reference, which is a review, has been cited in the Introduction, para.3 along with the genes.

The review presents nine mitochondrial genes which the authors divide into two groups, 1) Genes expressed in mitochondria (SLC25A1, TXNRD2, MRPL40, PRODH, and COMT) and 2) Genes that have been shown to have an impact on mitochondrial function (TANGO2, ZDHHC8, UFD1L, and DGCR8). In the abstract they mention only eight genes, the ninth gene COMT is mentioned in the text.

Revision Plan

Reviewer #2 (Significance (Required)):

The manuscript titled "Tbx1 haploinsufficiency causes brain metabolic and behavioral anomalies in adult mice which are corrected by vitamin B12 treatment" by Caterini et al. presents a comparative metabolomics study in the brains of mouse models carrying a heterozygous mutation in the transcription factor Tbx1. This mutation is contrasted with a chromosomal deficiency encompassing Tbx1, among other gene loci, known as Df1/+, which serves as a mouse model for 22q11 microdeletion syndrome. The primary and most significant finding of the study is that Tbx1 heterozygosity alone induces broad metabolomic alterations in the entire brain parenchyma, despite Tbx1 expression being confined to vascular endothelial cells. The authors leverage this observation to investigate the effects of dietary supplementation with vitamin B12, which alters the metabolome in a manner interpreted by the authors as rescuing or reversing the Tbx1 heterozygosity phenotype. This study holds promise for understanding the individual gene contributions to the penetrant behavioral phenotypes observed in Df1/+ and 22q11 affected subjects. This potential arises from the clear and consequential metabolic phenotypes described, notably the accumulation of methylmalonic acid.

However, despite the intriguing metabolic phenotypes, there are significant issues hindering incontrovertible conclusions.

Response to Reviewer #2

Major comments

1. Despite the intriguing metabolic phenotypes, there are significant issues hindering incontrovertible conclusions. Chief among these problems is the experimental design's nature, where the effects of genotype and a pharmacological intervention, vitamin B12, are assessed. The current design overlooks the effects of vitamin B12 on wild-type animals in metabolic and behavioral measures, thus precluding the attribution of the effects of vitamin B12 to a rescue.

See response to Reviewer 1 (point 2) who made the same criticism. This group is now included in the data analysis of the relevant experiments.

An alternative explanation, consistent with the measurements, is that vitamin B12 modifies metabolites and transcripts irrespective of genotype. A suggestion of this possibility is the observed effect of B12 lowering glutamate levels in Tbx1 mutant tissue below those in wild-type brain tissue (Fig. 4C).

This might be true for some metabolites. Indeed, we found 5 metabolites that responded similarly to vB12 in both WT and Tbx1 +/- mice. In contrast, three metabolites responded to vB12 in both WT and Tbx1 +/- mice, but the response was more pronounced in Tbx1 +/- mice. Finally, a group of eight metabolites was altered exclusively in Tbx1 +/- mice after vB12 treatment, including inosine, glutamate and short-chain fatty acids (SCFAs), Figure 4 and Supplementary Figure 6. Thus, overall, our data suggest that with only a few exceptions, the metabolic response to vB12 treatment is genotype-dependent.

This experimental design issue is exacerbated by the multitude of analytes measured by metabolomics, all collectively assumed to change as part of a common genotype-B12 interaction mechanism. This interpretation is feasible only if none of the analytes were to respond to B12 in wild-type animals.

As specified above, the response to vB12 was genotype-dependent. The inclusion of the vB12-treated WT dataset should address this point.

2. A second major issue arises from the assertion that Tbx1 is exclusively expressed in mouse brain endothelial cells and not in brain parenchyma. A significant unresolved question is how a gene expressed solely in endothelial cells can alter the brain parenchyma metabolome and transcriptome. This issue remains unaddressed and is not sufficiently discussed. If this assertion holds true, then the observations bear great importance in understanding how Df1/+ causes brain phenotypes and, by extension, in human 22q11.

There are quite a lot of published data from the mouse demonstrating the brain endothelial-specific expression of Tbx1 and the lack of expression in other brain cell types. These include studies using reporter genes (Paylor, 2006; Cioffi, 2014), Tbx1^{Cre} based cell fate mapping (Cioffi, 2014, Cioffi, 2022) and single cell whole genome transcriptions (Ximerakis et al., 2019); (https://portals.broadinstitute.org/single_cell/study/aging-mouse-brain). All are cited in the manuscript.

Revision Plan

HOW the mutation of *Tbx1* alters the brain metabolome and transcriptome will be the object of future studies, Currently, we do not have any data. At the reviewer's request, we have extended the discussion of this point in the revised manuscript (Discussion, para. 4).

*In this vein, the authors should consider that *Tbx1* is not expressed in brain endothelial cells in humans and is minimally expressed in fetal astrocytes (see <https://brainrnaseq.org/>).*

<https://brainrnaseq.org> provides a tool to evaluate the gene expression in the fetal brain. The sequencing was performed on fetal human brain tissue after elective pregnancy termination (4wks–9wks, it is not clear). Our analysis focuses on adult mice, which may contribute to observed differences.

Moreover, Yi et al. (2010) generated a gene expression atlas of human embryogenesis spanning from 4 to 9 weeks of gestational age, revealing downregulation of *TBX1* during this timeframe. Conversely, in the normal adult human brain, *TBX1* expression is identified in endothelial cells, as indicated in "The Human Brain Cell Atlas v1.0" presented for visualization and data mining through the Chan Zuckerberg Initiative's CellxGene application, referring to the atlas ontology in Ding et al. (2016).

3. A third major concern pertains to the general poor quality of the figures. Many figures appear to be directly exported from the software used for data analysis without proper curation. They are inadequately labeled, lack color codes to clarify differences (e.g., volcano plots), feature lettering fonts that are difficult to discern, and have lettering panels placed in awkward positions. Fig. 1 would benefit by the addition of a pathway diagram showing which metabolites are changing. Figure tables/spreadsheets either have sheets labeled in Italian or are empty. Collectively, the manuscript needs more careful data curation and presentation.

Many of the figures and tables have been modified with respect to the original manuscript and issues of clarity and quality have been improved where necessary.

Other points for consideration are listed below.

- *The abstract results section does not mention the *Df1* mutants at all, and overall the description of the results should be improved*

Corrected

- *The abstract would benefit from defining *vB12* before using the abbreviation*

Corrected

- *The section of the Results describing MMA accumulation in the brain would benefit from*

- 1) *explaining the choice of 1 month of age for terminal experiments and the choice to use whole brains (are there particularly brain regions suspected to be affected?),*

The majority of animals were 2 months of age at sacrifice (age and sex of individual animals are indicated in Supplementary Table 1). Young adult mice were the object of the study for the reason described in the first paragraph of the Results section, namely "Human studies of brain metabolism have mainly been conducted on children and adolescent patients. Therefore, in order to determine whether similar anomalies were present in the mouse models, we performed our studies on young mice between 1 and 2 months of age (Dutta et al., 2016)".

This is also the age at which the behavioural phenotype has been demonstrated (Paylor et al., 2006), and therefore could, potentially be rescued by *vB12* treatment. We do not have regional information pertaining to the adult brain.

- 2) *describing any sex effects for *Tbx1* mutants (and clarifying what data points for *Tbx1* animals correspond to which sex), and 3) including what sex was used for *Df1* experiments.*

In preliminary experiment we analyzed males and females' mice, before electing to use only males. To obtain reliable information about the impact of gender on metabolism and transcription we would have to use much larger numbers of animals. In Supplementary Table 1 pertaining to males and females are now indicated.

- *The authors demonstrate that *vB12* rescues PPI but use no other behavioral paradigms. It is possible that these mutations and/or *vB12* could be impacting anxiety-like behaviors or other behavioral phenotypes. By only including PPI, the authors limit the interpretation of the "rescue" of this phenotype by *vB12*.*

*Reduced PPI was the only behavioral anomaly identified in *Tbx1*^{+/-} mice that were subjected to a standard battery of behavioral tests (Paylor et al., 2006).*

Reduced PPI was the only behavioral anomaly identified in *Tbx1*^{+/-} mice that were subjected to a standard battery of behavioral tests (Paylor et al., 2006).

Reviewer #3 (Evidence, reproducibility and clarity (Required)):

*This is a terrific paper looking at influences of *Tbx1* heterozygosity on metabolic phenotypes in mice. A weakness is*

Revision Plan

that the locus on effects of B12 is totally unclear--could be neurovascular or even peripheral, but correcting this weakness might include study of Tbx1 conditional mutants, beyond the scope of this study.

*Reviewer #3 (Significance (Required)):
good significance*

Only two minor suggestions.

"We selected to study primarily Tbx1 single gene mutants because it is the primary candidate disease gene". What is the basis for this statement? Mouse +/- mutants in Mrpl40, Txnrd2, ProhD, and probably others have shown brain phenotypes.

The basis for TBX1 being considered as the primary candidate disease gene is the finding of TBX1 point mutations in patients who have the full spectrum of clinical phenotypes associated with 22q11.2 deletion syndrome without the chromosomal deletion, namely, congenital heart defects, immune defects, facial dysmorphism, learning defects and developmental delay. Similarly, in the mouse, Tbx1 mutation recapitulates the phenotype observed in multigene deletion mutants, such as Df1/+ mice.

We do not say (or think) that heterozygosity of other genes from del22q11.2 does not contribute to the disease, but mutations of other genes have not been found in individuals with a 22q11.2DS phenotype but without the chromosomal deletion.

In the discussion, the authors could close the loop on low glutamine could result in lower gaba in inhibitory interneurons, and its correction with B12 could restore gaba levels.

Discussion, para. 3. We thank the reviewer for comment. However, the GABA concentration is not altered in Tbx1 haploinsufficient brains; it is only upregulated by Vitamin B12. Therefore, this assumption may be very speculative. Due to differences in the release and reabsorption rates of the three compounds (glutamine, glutamate, and GABA), correctly evaluating the glutamine-glutamate cycle requires separating astrocytes from neurons. We have only discussed the upregulation of glutamate and the GABA response to Vitamin B12, which may counteract the excess of glutamate.

4. Description of analyses that authors prefer not to carry out

Please include a point-by-point response explaining why some of the requested data or additional analyses might not be necessary or cannot be provided within the scope of a revision. This can be due to time or resource limitations or in case of disagreement about the necessity of such additional data given the scope of the study. Please leave empty if not applicable.

Reviewer 1, point 11. The PPI decrease in the Tbx1+/- group appears to be driven by results from 3-4 mice. First, are those data statistical outliers?

Second, experiments in the same mice would be more informative. Do PBS-treated mutants recover PPI if they are treated with vB12 and vice versa? If the authors are concerned about the age difference, they also should include age-dependent effects on PPI.

We are unable to perform this experiment because, as stated in the manuscript, the mice were sacrificed at the end of the experiment and the brains preserved for histological analysis (not part of this study). The generation of mice for new experiments would take about one year. With all due respect, we do not believe that the data that would be obtained are sufficiently important to justify, ethically and economically, this work.

October 23, 2024

RE: Life Science Alliance Manuscript #LSA-2024-03075-T

Dr. Gabriella Lania
Institute of Genetics and Biophysics
Via Pietro Castellino 111
Napoli 80121
ITALY

Dear Dr. Lania,

Thank you for submitting your revised manuscript entitled "Tbx1 haploinsufficiency causes brain and behavior anomalies in adult mice, corrected by vitamin B12". We would be happy to publish your paper in Life Science Alliance pending final revisions necessary to meet our formatting guidelines.

- please address the Reviewer's remaining point
- please be sure that the authorship listing and order is correct
- please consult our manuscript preparation guidelines <https://www.life-science-alliance.org/manuscript-prep> and make sure your manuscript sections are in the correct order
- please add a Running Title, Category, and a Summary Blurb for your manuscript to our system
- please add the Twitter handle of your host institute/organization as well as your own or/and one of the authors in our system
- please use the [10 author names, et al.] format in your references (i.e. limit the author names to the first 10)
- please add a conflict of interest statement and the author contributions to the main manuscript text
- please add a figure callout for Figure S1 A and B, Figure S5 B, and Supplementary Figure 7 to your main manuscript text

A. FINAL FILES:

B. MANUSCRIPT ORGANIZATION AND FORMATTING:

Sincerely,

Reviewer #2 (Comments to the Authors (Required)):

The authors have done a great job addressing all my comments. The addition of the wt b12 treated group has turned the paper into a rigorous and interesting contribution with great potential for translation into humans. I would like to suggest that the authors review tables, in particular Fig. 2C as there are some spelling errors in the table.

November 7, 2024

RE: Life Science Alliance Manuscript #LSA-2024-03075-TR

Dr. Gabriella Lania
Institute of Genetics and Biophysics
Via Pietro Castellino
Naples 80131
Italy

Dear Dr. Lania,

Thank you for submitting your Research Article entitled "Brain and behavior anomalies caused by Tbx1 haploinsufficiency are corrected by vitamin B12". It is a pleasure to let you know that your manuscript is now accepted for publication in Life Science Alliance. Congratulations on this interesting work.

DISTRIBUTION OF MATERIALS:

Again, congratulations on a very nice paper. I hope you found the review process to be constructive and are pleased with how the manuscript was handled editorially. We look forward to future exciting submissions from your lab.

Sincerely,
